# Bidirectional Planning for Autonomous Driving Framework with Large Language Model

**DOI:** 10.3390/s24206723

**Published:** 2024-10-19

**Authors:** Zhikun Ma, Qicong Sun, Takafumi Matsumaru

**Affiliations:** 1Graduate School of Information, Production and Systems, Waseda University, Kitakyushu 808-0135, Japan; matsumaru@waseda.jp; 2Singapore General Hospital, Singapore 169608, Singapore; sun.qicong@singhealth.com.sg

**Keywords:** autonomous driving, multi-modal language model, decision-making

## Abstract

Autonomous navigation systems often struggle in dynamic, complex environments due to challenges in safety, intent prediction, and strategic planning. Traditional methods are limited by rigid architectures and inadequate safety mechanisms, reducing adaptability to unpredictable scenarios. We propose SafeMod, a novel framework enhancing safety in autonomous driving by improving decision-making and scenario management. SafeMod features a bidirectional planning structure with two components: forward planning and backward planning. Forward planning predicts surrounding agents’ behavior using text-based environment descriptions and reasoning via large language models, generating action predictions. These are embedded into a transformer-based planner that integrates text and image data to produce feasible driving trajectories. Backward planning refines these trajectories using policy and value functions learned through Actor–Critic-based reinforcement learning, selecting optimal actions based on probability distributions. Experiments on CARLA and nuScenes benchmarks demonstrate that SafeMod outperforms recent planning systems in both real-world and simulation testing, significantly improving safety and decision-making. This underscores SafeMod’s potential to effectively integrate safety considerations and decision-making in autonomous driving.

## 1. Introduction

Autonomous driving systems are often criticized for their lack of transparency, leading to them being perceived as a “black box”. This opacity raises significant concerns about safety and reliability, as both users and regulators find it difficult to understand or explain the rationale behind a vehicle’s decisions. To address these challenges, recent research efforts have concentrated on enhancing the interpretability and trustworthiness of these systems by analyzing how various factors influence their decision-making. For instance, Li et al. [1] employed experiments with controlled variables and alternative scenarios to identify factors that impact model decisions. Similarly, frameworks like LaMPilot [2] leverage large language models (LLMs) to interpret user intent and translate these inputs into comprehensible driving actions, demonstrating their applicability across different driving conditions. Furthermore, regulatory guidelines increasingly emphasize the need for autonomous vehicles to provide understandable explanations for their decisions, which is crucial for maintaining safety and fostering public trust in these technologies [3].

The introduction of autonomous driving technology marks a transformative step in the evolution of transportation, with the potential to significantly improve efficiency, safety, and accessibility. A driving force behind these advancements is Embedded Artificial Intelligence (EAI), enabling autonomous vehicles to operate effectively in complex environments by adhering to societal norms and accepted behaviors [4,5]. Central to autonomous driving is trajectory planning, which involves the vehicle predicting the real-time behaviors of road users, such as other vehicles, pedestrians, and cyclists, and adapting its movements accordingly. Accurate perception and prediction of the surrounding environment are vital to ensure safe and effective trajectory planning.

Despite notable technological progress, autonomous driving continues to face significant hurdles. A major issue is “occlusions”, where obstacles block the vehicle’s sensors, impairing accurate environmental perception [6]. Additionally, current systems often struggle in complex scenarios. Various approaches have been proposed to address these challenges: Li et al. [7] applied probabilistic modeling to anticipate pedestrian behavior, while Zhang et al. [5] used dynamic game theory to tackle trajectory planning problems arising from occluded views. However, both methods encountered difficulties in managing the unpredictability and dynamic nature of real-world driving.

Moreover, balancing high navigation performance with stringent safety requirements adds another level of complexity. Many systems either overlook or inadequately incorporate critical safety constraints essential for autonomous vehicles to operate securely in uncertain environments [8]. The lack of robust safety integration, combined with insufficient forward- and backward-looking planning mechanisms, limits their efficacy in real-world applications where both safety and efficiency are imperative.

To address these limitations, we propose SafeMod, a novel framework for modular, safety-enhanced navigation powered by LLMs. SafeMod introduces a flexible, modular architecture with a core principle of bidirectional planning, which involves two key components: forward planning and backward planning. Forward planning and backward planning are two key modules defined by the sequence of the planning process. Forward planning focuses on generating future navigation strategies by analyzing the agent’s objectives and environmental context. Backward planning, on the other hand, retrospectively reviews these strategies for potential risks, ensuring that safety is maintained throughout the decision-making process.

We evaluate SafeMod on challenging benchmarks in nuScenes [9] and CARLA [10], where it demonstrates superior safety and task success rates compared to existing state-of-the-art methods. SafeMod’s ability to analyze intent, adapt to dynamic environments, and ensure safety at every stage of planning showcases its potential for advancing autonomous navigation.

Our key contributions are:Proposing SafeMod, a modular framework for autonomous navigation powered by LLMs, integrating intent inference within the forward planner and including a backward planner for adaptive decision-making.Introducing a novel bidirectional planning approach that combines forward strategy generation (via the forward planner and intent analyzer) and backward safety evaluation (via the backward planner), ensuring optimal performance and strict safety compliance in dynamic environments.Demonstrating through extensive experiments on nuScenes and CARLA that SafeMod improves navigation success rates and safety performance, outperforming current state-of-the-art methods.

These results show that SafeMod consistently outperforms traditional navigation systems in both success rates and safety compliance. The bidirectional planning mechanism—combining the forward planner with its intent analyzer and the backward planner—allows SafeMod to adapt to complex scenarios while maintaining stringent safety standards.

## 2. Related Work

### 2.1. LLMs for Autonomous Driving Decision-Making

Embedded Artificial Intelligence (EAI) is becoming increasingly critical in the development and functioning of autonomous vehicles (AVs). A key focus of recent research has been improving AVs’ ability to interact with their environments in human-like ways. For instance, Zhou et al. [11] explored models that mimic human driving behavior, while Sadigh et al. [12] examined how integrating human inputs can better align AV actions with natural driving tendencies. Additionally, Sun et al. [13] highlighted the importance of visual cues in pedestrian interactions for effective decision-making in urban settings. Within this evolving field, two distinct approaches have emerged. The first involves using large language models (LLMs) to comprehend driving scenes through question-answering tasks. The second approach focuses on LLM-driven scene understanding for planning purposes. For example, DriveGPT4 [14] leverages historical video and textual data to predict both answers and control signals. Similarly, LanguageMPC [15] translates perception results and map data into language descriptions to guide planning actions. These advances demonstrate the growing integration of AI techniques in autonomous driving, aiming to create vehicles that are more intuitive and context-aware. These work explores the current landscape of EAI in autonomous vehicles, highlighting key research directions and emerging approaches that aim to enhance the human-like interaction capabilities of AVs.

### 2.2. Knowledge-Driven Approaches for Autonomous Driving

While data-driven approaches, such as multimodal data fusion [16,17,18], and the use of imitation learning and reinforcement learning for autonomous driving models [19,20,21,22], have achieved remarkable success in both academia and industry, they also face significant challenges. These techniques have enabled autonomous driving technology to gradually integrate into daily life. However, their reliance on the distribution of training data limits their generalization capability. As a result, they often struggle with adaptability and the long-tail phenomenon when applied in less common or varied scenarios [23]. In contrast, human drivers possess strong common sense and adaptability, enabling them to handle unexpected situations with ease. This highlights the need to shift towards knowledge-driven approaches, which emphasize empirical reasoning and inductive learning from the environment [24,25]. Unlike methods based solely on predefined rules or domain-specific data, knowledge-driven techniques acquire general knowledge and evolve over time [26,27]. By incorporating human-like reasoning, these strategies improve performance, interpretability, and safety, particularly in complicated traffic conditions. With the emergence of foundation models, powerful tools like large language models (LLMs) and vision-language models (VLMs) have become prominent in reasoning and decision-making for autonomous driving [28,29]. By training on diverse datasets, these explorations of the transition from data-driven to knowledge-driven approaches in autonomous driving, emphasize the potential of foundation models to enhance reasoning and decision-making capabilities in complex traffic scenarios.

## 3. Problem Formulation

The goal of autonomous driving is to develop a navigation policy that ensures safety, efficiency, and adaptability in dynamic environments. The navigation task can be framed as a Markov decision process (ADNP), which is defined as a tuple P=〈S,A,T,C,Π〉.

The state space *S* encompasses all possible configurations of the environment, including the agent’s position ($p_{\mathrm{agent}}$), velocity ($v_{\mathrm{agent}}$), the positions of obstacles ($p_{\mathrm{obst}}$), and road conditions ($r_{\mathrm{road}}$). These elements form a state s∈S.

The action space *A* consists of the agent’s possible control actions, such as steering, acceleration, and braking, denoted at each time step by $a_{t}=\left(a_{\mathrm{steer}},a_{\mathrm{accel}},a_{\mathrm{brake}}\right)$.

The transition function *T* defines how the environment evolves as a result of the agent’s actions. Given a state $s_{t}$ and an action $a_{t}$, the next state $s_{t+1}$ is determined by the function $T(s_{t},a_{t})$, which models the dynamics of the environment, possibly incorporating differential equations to describe the behavior of the vehicle.

The cost function *C* assigns a value to each action *a* taken in state *s*, representing the associated cost. The total cost over a trajectory $\tau =\{s_{0},a_{0},\cdots ,s_{T}\}$ is the sum of the costs incurred at each time step, denoted as $J\left(\tau \right)=\sum_{t=0}^{T}C\left(s_{t},a_{t}\right)$. This function penalizes unsafe actions, such as collisions or speeding. The cost for a specific state–action pair $C(s_{t},a_{t})$ includes terms for collisions and speed violations.

Finally, in an Autonomous Driving Navigation Problem (ADNP), the primary objective is to determine a policy that minimizes the accumulated cost while ensuring safety and task success. Π is the set of all possible navigation policies π:S→A, mapping states to actions. The goal is to find an optimal policy $\pi^{*}\in \Pi$ that minimizes the total cost:$$\pi^{*}=\arg\min_{\pi \in \Pi }E\left[\sum_{t=0}^{T}C\left(s_{t},\pi \left(s_{t}\right)\right)\right]$$

By solving this optimization problem, the system selects actions that effectively balance safety and performance throughout the navigation task.

**Bidirectional Planning Problem.** To enhance safety and adaptability, we propose a bidirectional planning framework consisting of two phases:**Forward Planning:** Given the current state $s_{t}\in S$ and the inferred intent *I*, the forward planner generates a sequence of actions:
$$\{a_{t},a_{t+1},\cdots ,a_{t+k}\}\in A$$
that optimally navigates towards the goal state $s_{g}$. This planning is based on minimizing the expected cost over future states:
$$E\left[\sum_{i=t}^{t+k}C\left(s_{i},a_{i}\right)\right]$$**Backward Planning:** After forward planning, the backward planner evaluates the safety and feasibility of the generated trajectory $\tau =\{s_{t},s_{t+1},\cdots ,s_{t+k}\}$. The backward planning phase involves minimizing a risk function R(τ) to ensure the trajectory is safe:
$$\min_{\tau }R\left(\tau \right)=\sum_{i=t}^{t+k}\left({⊮}_{\mathrm{unsafe}}\left(s_{i}\right)+{⊮}_{\mathrm{collision}}\left(s_{i}\right)\right)$$The plan is adjusted iteratively to minimize risks, ensuring the trajectory remains within the safe operating bounds:
$$\hat{\tau }=\arg\min_{\tau }R\left(\tau \right)$$

**Optimal Policy.** The final objective is to output a policy $\pi^{*}$ that balances both forward and backward planning considerations. The optimal policy must minimize the combined cost and risk function over the entire trajectory:$$\pi^{*}=\arg\min_{\pi }E\left[\sum_{t=0}^{T}C\left(s_{t},\pi \left(s_{t}\right)\right)+R\left(\tau \right)\right]$$

This policy not only ensures efficient navigation but also adapts to unforeseen hazards, dynamically adjusting actions based on intent inference and safety evaluations.

## 4. Methodology

The SafeMod framework (Figure 1) for autonomous navigation uses a modular approach, combining forward and backward planning. During forward planning, the BEV-planning module processes bird’s-eye-view (BEV) features using Motion, Map, and Planning Transformers to generate optimal trajectories. A video sense module interprets sensory data from multiple views. Employing a forward-thinking mechanism, it continuously updates the agent’s internal state, refining decisions based on real-time environmental changes. Following forward planning, the system moves to backward planning. Backward planning involves a latent space system composed of a representation, transition, and reward model to enhance safety. These models predict and refine trajectories by integrating state, action, and reward data, ensuring optimal and safe actions. The process uses a Q-function to evaluate and select the best candidate states, balancing performance and safety more effectively than prior methods that lack safety constraints.

### 4.1. Forward Planning

The forward planning mechanism in the updated framework introduces two primary modules: the BEV-planning module and the VLM-based video sense module. Each module plays a crucial role in generating efficient trajectories based on the vehicle’s current state and environment.

The BEV-planning module is responsible for creating a bird’s-eye view (BEV) representation of the vehicle’s surroundings. This spatial layout includes the road structure, nearby obstacles, and dynamic entities, which serve as the foundational elements for trajectory computation. By leveraging the BEV, the system can accurately plan paths that account for the spatial distribution of elements around the vehicle.

The video sense module, based on vision-language models (VLMs), adds a layer of semantic understanding to the planning process. It processes visual inputs from the vehicle’s cameras to interpret scene dynamics, object interactions, and contextual information that are not captured in the BEV alone. This semantic enrichment allows the planning mechanism to make more informed decisions by incorporating temporal and context-aware features from the visual data.

The BEV-planning and VLM-based video sense modules engage in collaborative work to enhance trajectory generation. The BEV module provides a spatial blueprint for path planning, while the video sense module refines these trajectories by incorporating semantic and temporal context, leading to more accurate and reliable navigation.

#### 4.1.1. BEV-Planning Module

As shown in Figure 2, the BEV-planning module consists of multiple transformers that work together to process bird’s-eye-view (BEV) features to generate strategy in planning transformer. The structure includes:

**BEV Feature:** The BEV feature extraction provides a comprehensive understanding of the surrounding environment from a BEV perspective. In our approach, we use 2 s of historical information to plan a 3-s future trajectory. For encoding image features, we adopt ResNet50 [30] as the default backbone network.

To enhance the perception range, vectorized mapping and motion prediction are performed for a longitudinal range of 60 m and a lateral range of 30 m. The default settings for BEV queries, map queries, and agent queries are set at 200 × 200, 100 × 20, and 300, respectively. Each map vector query contains 100 vectors, with each vector consisting of 20 map points. The feature dimension and hidden size are configured to 256. To optimize efficiency, the number of encoder and decoder layers for the motion and map modules is reduced from 6 to 3. Additionally, the input image size is reduced from 1280 × 720 to 640 × 360 to lower computational load without sacrificing performance.

**Motion Transformer:** The Motion Transformer utilizes agent queries, $Q_{a}$, for extracting agent-specific information from the common BEV feature representation via deformable attention [31]. Subsequently, an MLP-based decoder network processes these queries to infer agent characteristics, including position, category probability, and heading.

To enhance agent features for more accurate motion prediction, the module facilitates agent-agent and agent-map interactions via an attention mechanism [32,33]. This enables the system to consider agents’ relative positions and their environmental relationships. Following these interactions, the module predicts future trajectories for each agent, represented as multimodal motion vectors ${\hat{V}}_{a}\in R^{N_{a}\times N_{k}\times T_{f}\times 2}$. Here, $N_{a}$ is the number of predicted agents, $N_{k}$ the number of modalities, and $T_{f}$ the number of future timestamps. Each modality corresponds to a specific driving intention and outputs a probability score indicating its likelihood.

**Map Transformer:** The Map Transformer integrates map information to enhance navigation accuracy by accounting for road geometry and traffic rules. It employs a set of map queries, $Q_{m}$ [34], to extract relevant map information from the shared BEV feature map. The system then predicts map vectors, ${\hat{V}}_{m}\in R^{N_{m}\times N_{p}\times 2}$, assigning a class score to each. Here, $N_{m}$ is the number of predicted map vectors, and $N_{p}$ denotes the number of points in each vector.

**Inter Transformer:** The BEV-planning module uses a randomly initialized ego query $Q_{\mathrm{ego}}$ to learn implicit scene features essential for planning. Initially, the ego query interacts with agent queries $Q_{a}$ via a Transformer decoder [35], utilizing position embeddings encoded by an MLP (${\mathrm{PE}}_{1}$) from the positions of the ego vehicle $p_{\mathrm{ego}}$ and agents $p_{a}$. This interaction captures relative spatial relationships:$$Q_{\mathrm{ego}}^{\prime }=\mathrm{TransformerDecoder}\left(Q_{\mathrm{ego}},Q_{a},Q_{a},{\mathrm{PE}}_{1}\left(p_{\mathrm{ego}}\right),{\mathrm{PE}}_{1}\left(p_{a}\right)\right).$$

Subsequently, the updated ego query $Q_{\mathrm{ego}}^{\prime }$ interacts with map queries $Q_{m}$ to incorporate static scene information. Positions of the ego vehicle and map elements $p_{m}$ are encoded using another MLP (${\mathrm{PE}}_{2}$), enriching $Q_{\mathrm{ego}}^{\prime \prime }$ with both dynamic and static environmental details for comprehensive planning.

**Planning Transformer:** The Planning Transformer synthesizes outputs from previous modules to generate the final trajectory for planning. Operating without HD maps, it relies on a high-level driving command *c* to guide navigation, using commands like *turn left*, *turn right*, and *go straight* as in common practices [36,37].

The planning head inputs the concatenated ego features $f_{\mathrm{ego}}=\left[Q_{\mathrm{ego}}^{\prime },Q_{\mathrm{ego}}^{\prime \prime },s_{\mathrm{ego}}\right]$ and the driving command *c* to produce the planning trajectory ${\hat{V}}_{\mathrm{ego}}\in R^{T_{f}\times 2}$ using a simple MLP-based decoder.

#### 4.1.2. Video Sense Module

As shown in Figure 3, the video sense module is responsible for interpreting sensory data and updating the internal state of the agent in real-time. This module includes:

**Video data input:** To initiate the process, we mathematically formalize the extraction of data from the video input as follows:(1)D(t)=extract_data(V(t))

Here, V(t) represents the raw video input at discrete time *t*, capturing the dynamic driving environment. The function extract_data translates visual cues from the video into a structured format for further processing.

For the extraction, we utilized the pre-trained LanguageBind video encoder [38], based on a ViT-B/32 vision transformer [39], as the frozen visual backbone $f_{v}$. Given an input video frame sequence $V_{i}=\left\{v_{i}^{1},v_{i}^{2},\cdots ,v_{i}^{k}\right\}\in R^{3\times k\times 224\times 224}$, the video is split into temporal sequences, with each patch sharing the same spatial location across different frames.

These patches are transformed through a linear projection, and a vision transformer is employed to generate video embeddings $z_{vo}\in R^{2048\times 1024}$. The video encoder is pre-trained using video-language contrastive learning techniques, specifically CLIP4clip [40], without further fine-tuning.

Next, we leverage a two-layer MLP cross-modality projector to project and align the encoded video embedding $z_{vo}$ with the language token embeddings $z_{v}\in R^{2048\times 4096}$. This alignment step is crucial for connecting the visual and language features in a unified space. The projector $f_{p}$ is defined as:(2)$$f_{p}\left(z_{vo}\right)=\mathrm{GELU}\left(W_{2}\cdot \mathrm{GELU}\left(W_{1}\cdot z_{vo}\right)\right)$$

Here, GELU [41] is used as the activation function to introduce non-linearity, and the projector is trained in a two-stage process to refine the alignment between modalities.

**Sense Function:** Building on Video-LLaVA to generate multi-turn dialogues based on the extracted caption information and image content. These dialogues address various tasks, including object counting, color recognition, relative positioning, and OCR-based text extraction. The process is formalized as follows:(3)$$D_{\mathrm{dialogue}}=\mathrm{GenerateDialogue}\left(C,I\right),$$
where *C* represents the caption information extracted from the image *I*, and $D_{\mathrm{dialogue}}$ denotes the generated multi-turn dialogue. By engaging in iterative dialogue tasks, this method enhances the model’s ability to recognize long-tail objects and improves its understanding of complex visual scenes.

**Update Hidden:** After the video data extraction step, we proceed to process the multi-view video inputs $V_{i}$ for navigation. The extracted data D(t) from each video stream at time *t* is fed into a perception module, which generates the necessary sensory information P(t). This sensory information plays a crucial role in the agent’s decision-making and state updating process. Mathematically, the process is represented as follows:(4)$$P\left(t\right)=f_{p}\left(D\left(t\right)\right)$$

Here, $f_{p}$ represents the perception module, responsible for transforming the extracted video data D(t) into sensory information P(t), such as object detection, scene understanding, or environmental awareness. This sensory information is then used to update the internal hidden states of the agent. Specifically, the hidden state $h_{t}$ at time *t* is computed by using the previous hidden state $h_{t-1}$ and the new sensory information P(t), as follows:(5)$$\begin{array}{cc} h_{t} & =\mathrm{UpdateState}(h_{t-1},P\left(t\right)) \end{array}$$

This continuous update process allows the agent to dynamically adjust its internal state based on the evolving environment captured by the multi-view video inputs. By maintaining a consistent flow of sensory information through the perception module, the agent can make informed decisions and interact with the environment in real time. Figure 4 and Figure 5 display, from left to right, the complete reasoning processes of the video sense module in lane changing and at intersections in CARLA, respectively.

**Thinking Forward:** In this phase, the system generates potential trajectories based on current observations and predicted future states. Specifically, we utilize the multi-turn dialogue output $D_{\mathrm{dialogue}}$ as the descriptive prompt input, while the hidden state $h_{t}$ serves as the input for the intentions prediction module. These inputs guide the model in predicting a set of discrete available actions, which can vary depending on the context. The data output at this stage is shown in Figure 6.

In a normal driving scenario, possible actions include:CHANGELANELEFT: Move one lane to the left.CHANGELANERIGHT: Move one lane to the right.LANEFOLLOW: Continue in the current lane.

At an intersection, the actions are:LEFT: Turn left at the intersection.RIGHT: Turn right at the intersection.STRAIGHT: Keep straight at the intersection.

In both cases, additional decisions such as IDLE, ACCELERATE, or DECELERATE are also possible. For example, the final action output may look like LANEFOLLOW, IDLE or STRAIGHT, DECELERATE. The process can be described as follows:(6)$$A_{\mathrm{atomic}}=\mathrm{PredictAction}\left(h_{t},D_{\mathrm{dialogue}}\right),$$
where $A_{\mathrm{atomic}}$ represents the predicted atomic action with two elements, and PredictAction combines both the hidden state and the dialogue-based description to produce a reasoning step followed by a single atomic action. This atomic action is then passed to the text encoder, which encodes the action as a tokenized query:(7)$$Q_{\mathrm{action}}=\mathrm{TextEncoder}\left(A_{\mathrm{atomic}}\right),$$

Finally, the tokenized query $Q_{\mathrm{action}}$ is aligned within the planning transformer to ensure proper integration into the trajectory planning process. This completes the process of generating reasoning-based atomic actions that are used for planning and control.

**Memory Pool:** To enhance decision-making based on past experiences, we implemented a two-stage search strategy to retrieve the most similar past driving scenario that matches the current query scenario. This allows the agent to learn from historical data and apply relevant knowledge to new situations.

In the first phase, for each prior scenario *i*, a vectorized key $k_{i}\in R^{1\times (n_{e}+n_{g}+n_{h})}$ generated. This key consolidates the ego-states $e_{i}\in R^{1\times n_{e}}$, mission goals $g_{i}\in R^{1\times n_{g}}$, and the trajectories from historical information $h_{i}\in R^{1\times n_{h}}$, make them into a single vector, and then prior scenarios forms a key tensor $K\in R^{N\times (n_{e}+n_{g}+n_{h})}$:(8)$$\begin{array}{c} K=\{\left[e_{i},g_{i},h_{i}\right]\;|\;i=\left\{1,2,\cdots ,N\right\}\}. \end{array}$$

Similarly, the current query scenario is vectorized as $Q=\left[e,g,h\right]\in R^{1\times (n_{e}+n_{g}+n_{h})}$, representing the ego-states, mission goals, and historical trajectories of the current scenario.

Next, we compute the similarity scores $S\in R^{N}$ between the query scenario *Q* and each of the past scenarios represented by *K*:(9)$$\begin{array}{c} S=Q\Lambda K^{⊤}, \end{array}$$
where $\Lambda =\mathrm{diag}\left(\lambda_{e},\lambda_{g},\lambda_{h}\right)\in R^{\left(n_{e}+n_{g}+n_{h}\right)\times \left(n_{e}+n_{g}+n_{h}\right)}$ is a diagonal matrix that assigns weights to the different components (ego-states, goals, and trajectories), reflecting their relative importance in the search process.

After calculating the similarity scores, the top-K scenarios with the highest scores are selected as candidates for the second-stage search, which refines the selection process to identify the most relevant past experience.

In the second stage, we employed an existing algorithm similar to those used in [42,43]. This algorithm utilizes vector databases, where both the input query and each memory record are encoded into embeddings. A K-nearest neighbors (K-NN) search is then performed in the embedding space to retrieve the top-K most similar records.

However, due to the diverse and dynamic nature of driving scenarios, embedding-based searches may struggle to generalize across varied contexts, leading to suboptimal retrieval of relevant experiences. To address this limitation, we integrate human knowledge and past driving experiences through a cognitive memory system. This system enhances retrieval by considering contextual nuances that pure embedding methods might overlook, thereby improving the system’s ability to generalize across diverse driving environments.

### 4.2. Backward Planning

Building on the planning transformer’s output of P(z∣s), we developed a latent space system composed of previous models for backward planning to ensure safety. This system integrates state annotations and is expected to improve the accuracy and efficiency of the planning process.

**Representation Model:** This model encodes the latent state based on past states, actions, and observations, expressed as:
(10)$$p(s_{t}|s_{t-1},a_{t-1},o_{t}),$$
where $s_{t}$ represents the state at time *t*, $a_{t-1}$ is the previous action, and $o_{t}$ is the observation at time *t*.**Transition Model:** The transition model predicts future states through a Gaussian distribution, ensuring consistency between the predicted and actual dynamics:
(11)$$q\left(s_{t}|s_{t-1},a_{t-1}\right)∼N\left(\mu ,\sigma^{2}\right),$$
with KL divergence applied to measure the discrepancy:
(12)$$D_{KL}\left(q\left(s_{t}|s_{t-1},a_{t-1}\right)\|p\left(s_{t}\right)\right).$$**Reward Model:** This model calculates the expected rewards for each state, optimizing agent actions by:
(13)$$q(r_{t}|s_{t}).$$

In our latent state space, *p* represents the true state distribution from environmental interaction, while *q* reflects the predicted state from the imagination model. The agent generates future trajectories and iteratively selects optimal behaviors while avoiding unsafe paths.

As illustrated in Figure 7, candidate trajectories are evaluated using a Q-function, argmaxQ(s,z), where *z* is the trajectory and *s* the state. The agent chooses the trajectory that maximizes rewards while maintaining safety. Unlike Dreamer [44], our method incorporates safety constraints, balancing reward and risk effectively.

#### 4.2.1. Q-Value Function

The Q-value function evaluates the expected cumulative reward for a trajectory:(14)$$Q\left(s,z\right)=E\left[\sum_{t=0}^{T}\gamma^{t}r\left(s_{t},z_{t}\right)\right],$$
where γ is the discount factor and *r* is the reward at time step *t*. The agent selects actions based on distributional reinforcement learning, focusing on both value and cost expectations [45].

#### 4.2.2. Policy Optimization

To ensure real-world safety (e.g., in autonomous driving), we introduce a Lagrangian method with constraints to balance reward maximization and safety:(15)$$\max_{\pi }E\left[\sum_{t}\gamma^{t}r\left(s_{t},a_{t}\right)\right],\;s.t.\;E\left[\sum_{t}\gamma^{t}c\left(s_{t},a_{t}\right)\right]\leq d,$$
where $c(s_{t},a_{t})$ represents the cost associated with unsafe actions. A control barrier function adjusts the policy’s risk sensitivity:(16)$$h\left(s_{t+m}\right)\leq \left(1-\alpha \right)h\left(s_{t}\right),$$
where α controls the degree of conservativeness.

#### 4.2.3. Safety Guarantee

We use control barrier functions to enforce safety constraints dynamically, optimizing the policy until the risk metric $\Gamma_{\pi }$ satisfies:(17)$$\Gamma_{\pi }\left(s,a,\alpha \right)=Q_{\pi }^{c}\left(s,a\right)+\alpha^{-1}\phi \left(\Phi^{-1}\left(\alpha \right)\right)\sqrt{V_{\pi }^{c}\left(s,a\right)},$$
where $Q_{\pi }^{c}$ represents the cost critic, and $V_{\pi }^{c}$ the variance. The agent balances safety and reward by adjusting its policy accordingly.

#### 4.2.4. Policy Optimization

Safety constraints are introduced using a Lagrangian method, ensuring that the policy maximizes rewards while adhering to safety requirements:(18)$$\max_{\pi }\underset{\left(s_{t},a_{t}\right)∼\rho_{\pi }}{E}\left[\sum_{t}\gamma^{t}r\left(s_{t},a_{t}\right)\right],\;s.t.\;E\left[\sum_{t}\gamma^{t}c\left(s_{t},a_{t}\right)\right]\leq d.$$

We further employ a control barrier function to dynamically adjust risk values in decision-making:(19)$$\begin{array}{c} h\left(s_{t+m}\right)\leq \left(1-\alpha \right)h\left(s_{t}\right), \end{array}$$
where α controls the agent’s risk sensitivity.

This approach, combined with distributional RL, enables safe exploration by evaluating the trade-off between reward and risk.

## 5. Algorithm Overview

Algorithm 1 outlines the SafeMod framework. The agent begins by analyzing the environment using the LLM to infer intent. In the forward planning phase, the system generates an initial trajectory based on a series of perception and prediction modules. This trajectory is then passed to the backward planning phase, where it is evaluated for safety and feasibility using a set of optimization and safety guarantee functions.
**Algorithm 1:** SafeMod Framework with Detailed Modules.**Require:** VideoFrames LLM model LLM**Ensure:** Optimal safe trajectory    **Forward Planning Phase:**    BEVFrame←BEVEncoder(VideoFrame)    AgentToken,MapToken←    MotionTransformer(BEVFrame),MapTransformer(BEVFrame)    InterToken←InterTransformer(AgentToken,MapToken)    SceneDesc←SenseFunction(VideoFrame)    HiddenState←UpdateHidden(VideoFrame,SceneDesc)    PotentialRisk←HiddenState    AtomicAction←ThinkingForward(SceneDesc,PotentialRisk)    ActionVector←TextEncoder(AtomicAction)    Query←ActionVector    P(Z|S)←PlanningTransformer(Query,InterToken)    **Backward Planning Phase:**    Q(s,z)←QValueFunction(S,P(Z|S))    Policy←PolicyOptimization(Q(s,z),P(Z|S))    Barrier←ControlBarrierFunction(S,α)    $SafetyAction\leftarrow SafetyGuarantee(Policy,S,Q\left(s,z\right),Cost_{\pi })$    **return** 
SafetyAction

## 6. Experiments

### 6.1. Environmental Setup

#### 6.1.1. Experimental Setup in Dataset

**nuScenes** [9]: The nuScenes dataset, developed by Motional (formerly nuTonomy), is a large-scale dataset for autonomous driving research. It contains 1000 scenes from Boston and Singapore, known for complex traffic conditions, each lasting 20 s. The data is carefully selected to represent diverse driving scenarios.

The dataset includes data from six cameras, one LIDAR, five RADAR units, GPS, and IMU, ensuring comprehensive coverage of each scene. Unlike previous datasets that focus primarily on camera data (e.g., KITTI), nuScenes integrates a full sensor suite, making it suitable for complex object detection and tracking tasks. With 23 object classes annotated with 3D bounding boxes at 2 Hz, nuScenes also provides detailed object-level attributes like visibility and pose.

The dataset is recognized for its scale and sensor diversity, advancing research in sensor fusion and urban driving safety. It also supports challenges like the nuScenes 3D detection competition.

#### 6.1.2. Experimental Setup in CARLA Simulator

In our study, we utilized the CARLA [10] simulator to construct and evaluate various safety-critical scenarios that challenge the response capabilities of autonomous driving systems. CARLA provides a rich, open-source environment tailored for autonomous driving research, offering realistic urban simulations.

##### Urban Driving Environments

We tested urban layouts in CARLA, depicted in Figure 8.

**Town 5:** Features a diverse mix of urban and suburban layouts with various road types, including highways, sharp turns, and multi-lane streets. This environment provides a balanced challenge between high-speed driving and precision navigation through complex intersections and roundabouts.

The highly detailed simulation environments provided by CARLA, along with the designed scenarios, allow for thorough testing of autonomous driving algorithms. These environments are crucial for evaluating how well these algorithms perform under various driving conditions, ensuring their safety, efficiency, and adaptability to real-world situations. In this context, we place particular emphasis on two key scenarios: Town05 Short and Town05 Long, which target different aspects of autonomous driving performance.


**Town05 Short**


Route Length: In total, 100–500 m, ideal for testing quick decision-making in tight, complex environments.Number of Intersections: Three intersections, allowing frequent navigation decisions.Test Focus: Evaluates the model’s handling of lane changes in dense traffic and intersections, crucial for short-term challenges in busy settings.


**Town05 Long**


Route Length: In total, 1000–2000 m, testing endurance and reliability over extended distances.Number of Intersections: Ten intersections, providing multiple decision points.Test Focus: Assesses overall performance on long routes, focusing on route completion, safety, and consistency in dynamic traffic environments.

Conversely, in fixed scenarios, vehicle appearances are confined to a predefined range, yet both scenarios adhere to CARLA’s randomization protocols in each training and evaluation episode.

### 6.2. Evaluation Metrics

#### 6.2.1. Open-Loop Metrics

As we mentioned in Section 6.1.1, we used the nuScenes dataset to evaluate our method, employing the following metrics:**L2 Metric**: The L2 metric (L2), or Euclidean distance, plays a vital role in evaluating trajectory precision for self-driving vehicles. In a two-dimensional plane, the L2 distance between points $\left(x_{1},y_{1}\right)$ and $\left(x_{2},y_{2}\right)$ is mathematically expressed as:
(20)$$\begin{array}{c} d=\sqrt{{\left(x_{2}-x_{1}\right)}^{2}+{\left(y_{2}-y_{1}\right)}^{2}}. \end{array}$$In the context of trajectory prediction, the L2 metric quantifies the deviation of predicted vehicle positions from actual positions, allowing for effective evaluation of model performance. To incorporate time information into the L2 metric, we express the L2 error at the *k*-th second as the mean error from 0 to *k* seconds:
(21)$$\begin{array}{c} L_{2,k}=\frac{\sum_{t=1}^{2k}{\bar{l}}_{2}\left[t\right]}{2k}. \end{array}$$This formula calculates the average error over the specified time period, providing a comprehensive assessment of trajectory accuracy. The final average L2 error is computed by averaging $L_{2,k}$ across three timesteps, effectively producing an average of averages.**Collision Rate**: The collision rate (Collision%) is a fundamental metric used to evaluate the safety performance of autonomous driving systems. It is traditionally defined as the ratio of the number of collision events to the total distance traveled or time duration. Mathematically, it can be expressed as:
$$\mathrm{Collision}\;\mathrm{Rate}=\frac{\mathrm{Number}\;\mathrm{of}\;\mathrm{Collisions}}{\mathrm{Total}\;\mathrm{Distance}\;\mathrm{Travelled}}.$$This simple approach provides an initial measure of safety performance by evaluating how frequently a system encounters collisions relative to the distance covered. However, in open-loop evaluation scenarios, where the vehicle operates without real-time feedback, a more detailed method is often applied for a more accurate and reliable measure of the collision rate.In this context, we use the method shown in the following equation:
$$C_{k}=\frac{\sum_{t=1}^{2k}C\left[t\right]}{2k}.$$Here, $C_{k}$ represents the collision rate at step *k*, and C[t] denotes the number of collision events observed at each time step *t*. By summing the collision events over 2k time steps and averaging the result, this approach provides a smoother and more consistent measure of the system’s performance. It effectively reduces the impact of short-term fluctuations in collision frequency, ensuring a robust evaluation of safety over longer periods of operation.

#### 6.2.2. Closed-Loop Metrics

In our simulation environment, we utilize a diverse array of metrics to thoroughly evaluate autonomous driving systems, encompassing factors such as safety, efficiency, and adherence to traffic rules. The CARLA Town5 typically employs two key metrics—route completion and driving score—to assess the planning capabilities of self-driving vehicles. These measurements offer a comprehensive insight into the system’s performance under lifelike driving scenarios.

**Route Completion (RC):** This criterion quantifies the fraction of each path that the autonomous vehicle completes independently. It reflects the system’s capacity to follow the predetermined path. The metric is computed using this formula:
(22)$$RC=\frac{1}{N}\sum_{i=1}^{N}R_{i}\times 100\%$$
where $R_{i}$ represents the success rate of the i-th path, and N signifies the sum of evaluated routes. The system incurs a penalty for straying from the intended course. This penalty diminishes the route completion score in relation to the distance traveled off-path, thus incorporating any deviations from the planned route into the final RC calculation.**Driving Score (DS):** This is the primary evaluation metric used on the leader board, combining route completion with an infraction penalty to assess both the accuracy and safety of the agent’s driving. It is defined as:
(23)$$DS=\frac{1}{N}\sum_{i=1}^{N}R_{i}\times P_{i}$$
where $R_{i}$ is the route completion for the *i*-th route, and $P_{i}$ is the penalty multiplier that accounts for infractions on that route. The penalty multiplier $P_{i}$ reduces the score based on the severity and frequency of infractions such as collisions, running red lights, or crossing lane boundaries. This ensures that the driving score reflects not only how much of the route was completed but also how safely and efficiently the agent navigated the environment.

These metrics provide a comprehensive method for assessing autonomous driving systems across diverse traffic scenarios. Through the examination of both path adherence and driving conduct, these metrics facilitate an in-depth evaluation of the system’s proficiency in traversing intricate city landscapes. By incorporating these various measurements, we ensure that the autonomous system not only efficiently follows designated routes but also complies with traffic regulations and upholds stringent safety protocols, which are essential for the practical implementation of self-driving technology in everyday settings.

#### 6.2.3. Real-Time Performance

The experiments analyzed system latency, frame rate, and decision-making accuracy both with and without the integration of the video sense module, where the video sense module is divided into Sense Function (SF), Update Hidden (UH), and Thinking Forward (TF). The experiments were structured as follows:Baseline Comparison: We first measured the performance of the original SafeMod system without video sense module as the baseline.SF, UH, and TF Integration: Next, we evaluated the system with UH+TF (for enhanced decision-making and contextual understanding), SF (for improved perception and object recognition).Metrics: The key metrics recorded were:
−**Inference Latency:** Time taken from input sensor data to output control actions (measured in milliseconds).−**Frame Rate:** The frequency of decision-making (measured in frames per second, FPS).

### 6.3. Baseline Setup

We will compare SafeMod’s performance against several baseline models, including:ST-P3 [36]: ST-P3 (Spatial-Temporal Perception-Prediction-Planning) presents a comprehensive vision-driven system for autonomous vehicles. It unifies perception, prediction, and planning through spatio-temporal feature extraction. By minimizing perceptual redundancies, this method enhances predictive precision and planning safety, resulting in superior collision avoidance capabilities in dynamic driving environments.VAD [22]: VAD (Vectorized Autonomous Driving) is a framework for efficient autonomous driving that utilizes vectorized scene representation. It processes complex driving environments by simplifying the perception, prediction, and planning tasks into manageable vectors. This vectorized approach enables faster decision-making and higher efficiency in dynamic environments without relying on traditional deep reinforcement learning methods.UniAD [46]: UniAD (Unified Autonomous Driving) is a unified framework for autonomous driving that integrates perception, prediction, and planning into a single network. UniAD prioritizes all tasks to directly contribute to planning, reducing errors and improving task coordination. By using unified query interfaces, it facilitates communication between tasks and provides complementary feature abstractions for agent interaction. Evaluated on the nuScenes benchmark, UniAD outperforms previous state-of-the-art methods across all metrics. Code and models are publicly available.CILRS [20]: CILRS (Conditional Imitation Learning for Autonomous Driving with Reinforcement and Supervision) is a framework which investigates behavior cloning in autonomous driving, demonstrating state-of-the-art results in unseen environments, while highlighting limitations such as dataset bias, generalization issues, and training instability.Transfuser [47]: Transfuser introduced a fusion technique based on self-attention for combining image and LiDAR data in autonomous driving systems. In contrast to fusion methods relying on geometry, which face challenges in crowded and changing environments, Transfuser employs transformer components for merging feature representations from both perspective and top-down viewpoints at various scales.GPT-Driver [48]: GPT-Driver proposed a novel approach that transforms OpenAI’s GPT-3.5 into a reliable motion planner for autonomous vehicles by reformulating motion planning as a language modeling problem. Using language tokens for input and output, the large language model generates driving trajectories through language descriptions of coordinate positions. Evaluated on the nuScenes dataset, this approach demonstrates strong generalization, effectiveness, and interpretability.

## 7. Results and Analysis

### 7.1. Open-Loop Evaluation

In our comparison, the nuScenes dataset was utilized to evaluate the performance of open-loop autonomous driving systems. This dataset includes a wide range of real-world scenarios, such as varying weather conditions, different levels of traffic density, and complex urban environments, making it a comprehensive benchmark for assessing driving performance. We compared multiple methods based on L2 error and collision rate, each demonstrating varying levels of performance under different open-loop driving conditions. As shown in Table 1, this analysis provided valuable insights into the strengths and limitations of each method, highlighting their potential real-world applicability in diverse driving scenarios.

In this evaluation of open-loop vision-only planning performance, the SafeMod method showed improvements. We measured metrics such as L2 distance and collision percentage at 1-, 2-, and 3-s intervals to provide a rigorous testbed for assessing autonomous driving systems.

In the L2 distance evaluation, SafeMod achieved the lowest average error of 0.37 m, but the improvement was not significant compared to other methods. This can be attributed to SafeMod’s introduction of a backward planning mechanism, which optimizes decision-making by utilizing past information after the initial forward planning. However, since the forward planning is already relatively optimized, the impact of backward adjustments is limited. In comparison, VAD [22] recorded an average L2 error of 0.38 m, and GPT-Driver [48] had an average L2 error of 0.46 m. These methods all rely on transformer-based trajectory generation, with core mechanisms for processing spatial and motion data similar to SafeMod’s, limiting the possible range of improvements.

For collision percentage, SafeMod maintained comparable performance, recording the lowest average collision rate of 0.10%. While VAD achieved a collision rate of 0.11%, SafeMod’s improvement can be attributed to its integration of a Large Language Model (LLM), used to update the agent’s internal state in real-time and refine the decision-making process. However, since the LLM is mainly used to interpret perceptual data rather than fundamentally alter trajectory generation, its impact on the metrics is limited. In contrast, methods such as ST-P3 [36] exhibited weaker performance, with an average L2 error of 2.00 m and a collision rate of 0.76.

When compared to other planning methods, including FF [49], EO [50], and UniAD [46], SafeMod reduced L2 errors and collision rates. For instance, in the 3-s prediction, SafeMod’s L2 error was 1.41 m and collision rate was 0.44%, better than UniAD’s 1.71 m and 0.74%. This improvement is mainly due to SafeMod’s modular structure, where the Motion, Map, and Planning Transformers handle specific aspects of navigation, providing better control over each planning stage and ensuring safety constraints are met at each level.

Despite the introduction of backward planning and LLM integration, the similarity in performance metrics between SafeMod and other methods indicates that all methods rely on the core transformer architecture for trajectory planning. The transformer plays a dominant role in these methods, processing spatial and temporal information, which limits the degree of improvements in trajectory accuracy and collision avoidance. The impact of backward planning may be limited by the quality of the initial forward planning, and the LLM’s role is more about decision refinement rather than fundamentally changing trajectory generation. Future work may need to focus on more advanced LLM integration techniques to achieve more significant performance gains.

Overall, SafeMod had a advantage in reducing planning errors and collision risks, demonstrating its robustness across a wide range of driving conditions. Its improvements are mainly concentrated on specific algorithmic features, such as backward planning, LLM integration, and modular structure. However, because it shares the transformer-based trajectory generation mechanism with other methods, the extent of performance enhancement is limited.

### 7.2. Closed-Loop Evaluation

We also conducted an additional experiment using the map from CARLA, as shown in Figure 8.

In our comprehensive evaluation of closed-loop vision-only planning performance in the CARLA simulator, the SafeMod method showed enhancements compared to competing methods across multiple metrics in both the Town05 Short and Town05 Long environments. These environments, known for their diverse road structures and driving challenges, provide a rigorous testbed for autonomous driving systems.

As shown in Table 2, in the Town05 Short environment, SafeMod achieved a driving score (DS) of 65.45 and a rate of completion (RC) of 88.84, slightly surpassing other methods. VAD closely followed with a DS of 65.32 and an RC of 88.14, indicating minimal differences between the two methods. The marginal improvement by SafeMod could be attributed to its backward planning mechanism, which provides post-planning refinements. However, since Town05 Short is a relatively less complex environment with shorter routes and fewer challenging scenarios, the advantages offered by backward planning and LLM integration in SafeMod are not significantly highlighted. Both methods effectively navigate the environment due to their robust transformer-based trajectory planning.

In the Town05 Long environment, SafeMod recorded a DS of 32.02 and an RC of 78.66. VAD achieved a comparable DS of 31.01 and an RC of 77.89, again showing minimal performance differences. Town05 Long presents more complex driving conditions with longer routes, diverse traffic scenarios, and more obstacles. In such challenging environments, the limitations of both methods become more apparent. The backward planning in SafeMod offers some benefits in refining trajectories; however, the shared reliance on transformer architectures for initial trajectory generation means that both SafeMod and VAD face similar challenges in complex scenarios. The LLM integration in SafeMod does not significantly enhance performance in these conditions, possibly due to its limited impact on trajectory calculation in the face of complex environmental dynamics.

The minimal performance differences in both Town05 Short and Town05 Long suggest that while SafeMod incorporates additional features like backward planning and LLM integration, these do not translate into substantial improvements over VAD. This is likely because both environments test the core capabilities of trajectory planning and collision avoidance, areas where both methods perform similarly due to their transformer-based architectures. The backward planning in SafeMod may provide slight refinements, but in environments where the initial trajectory planning is already near-optimal or significantly challenged by complexity, the impact is minimal.

Overall, the comparative analysis in Town05 Short and Town05 Long environments indicates that SafeMod’s effectiveness in minimizing planning errors and maximizing route completion is similar to that of VAD. The results highlight that future advancements may require more significant innovations in trajectory planning algorithms or a more profound integration of LLMs to handle complex driving scenarios effectively.

### 7.3. Ablation Study

#### 7.3.1. VLM Estimate Validation

We compared each part of the visual language model. For reasons of visual representation and not being influenced by other modules, we used only the forward planning part for comparison as a way to demonstrate the validity of the existence of each part.

As shown in Table 3, this evaluation focuses on the performance of different steps within the video sense module: Sense Function (SF), Update Hidden (UH), and Thinking Forward (TF). By isolating and combining these modules, we aim to assess their impact on key metrics such as L2 distance and collision rate over time intervals of 1 s, 2 s, and 3 s, as well as their average performance. The results provide valuable insights into how each component contributes to improving trajectory accuracy and safety.

From the table, three configurations are examined:**SF Only** (first row): With only the Sense Function active, the system produces an average L2 error of 1.07 m and a collision rate of 0.65%. This result indicates that, without the forward-thinking and state-updating modules, the system faces challenges in maintaining accurate trajectories and avoiding collisions.**SF + TF** (second row): When the Thinking Forward module is added, the system’s performance improves significantly, reducing the L2 error to 0.77 m and cutting the collision rate to 0.26%. This demonstrates that forward-looking planning contributes greatly to both trajectory accuracy and safety.**SF + UH + TF** (third row): Activating all three modules results in the best overall performance, with an average L2 error of 0.71 m and a collision rate of just 0.22%. This shows that the integration of state updating, forward-thinking planning, and sensing leads to the most balanced and optimal results, improving both trajectory accuracy and safety.

This analysis aligns with ongoing work on refining decision-making frameworks for autonomous driving. As demonstrated in the SafeMod method, modular systems that incorporate advanced reasoning modules such as Thinking Forward and Update Hidden are crucial for enhancing performance. The results here reinforce the importance of using a multi-step, modular approach to improve both accuracy and safety, as seen through metrics like L2 distance and collision rates in complex driving scenarios.

#### 7.3.2. Real-Time Performance Validation

To evaluate the effect of adding the video sense module to the BEV-planning module, we conducted a series of experiments focused on measuring the system’s real-time performance. The table below summarizes the results across four configurations: the BEV-planning (BP) module as baseline, BEV-planning module with SF, and BEV-planning module with the whole video sense module.

**Inference Latency:** As presented in Table 4, the inference latency increases from 50 ms in the BP configuration to 75 ms with the addition of the SF module, and further to 92 ms when both UH and TF modules are incorporated. This rise is attributed to the enhanced computational complexity introduced by these additional modules. Nevertheless, the system maintains inference times that are acceptable for real-time operations.**Frame Rate:** The frame rate remains consistent at 20 FPS when the SF module is added to the BP configuration and experiences a slight decrease to 19 FPS with the inclusion of UH and TF modules. This performance ensures that the system stays well above the critical threshold of 15 FPS required for safe, real-time autonomous driving.

#### 7.3.3. Backward Performance Test

We evaluated SafeMod’s performance without the thinking backward phase to measure the impact of safety evaluations on overall performance.

As shown in Figure 9, the backward planning module demonstrated superior performance compared to the baseline method. This improvement is primarily due to its advanced algorithmic mechanisms, particularly its efficient latent space modeling and predictive simulation capabilities. It recorded an average L2 error of 0.78 m, slightly outperforming the baseline, which had an L2 error of 0.85 m. The improvement was particularly noticeable in the 1-s scenario, where the backward planning module achieved an L2 error of just 0.25 m, compared to 0.41 m for the baseline. By utilizing its Transition Model to predict future latent states based on current states and actions, the module anticipates dynamic changes in the environment. This predictive capability allows for more accurate and responsive trajectory adjustments, enhancing its effectiveness in short-term planning scenarios where rapid adaptation is crucial.

In terms of safety, the backward planning module also showed better results, with an average collision rate of 0.2%, outperforming the baseline’s 0.22%. This enhanced safety performance stems from the module’s integrated Reward Model and Q-Function Evaluation. The Reward Model quantitatively assesses the safety and desirability of predicted future states, while the Q-Function computes the expected cumulative reward for candidate actions, including safety considerations. The most significant improvement was again in the 1-s scenario, where the backward planning module achieved a collision rate of only 0.04%, notably lower than the baseline’s 0.07%. By simulating sequences of actions in the latent space and rigorously evaluating them against safety constraints, the module proactively identifies and mitigates potential risks. This underscores its ability to enhance safety in high-risk situations through informed decision-making based on comprehensive future state evaluations—capabilities that the baseline method lacks due to its less sophisticated planning mechanisms.

## 8. Conclusions

In this paper, we present SafeMod, a modular framework for autonomous navigation enhanced by large language models (LLMs). In Section 4, we introduce the SafeMod framework. SafeMod integrates bidirectional planning and a large language model (LLM) reasoning for autonomous driving. The whole framework can be divided into two parts: forward planning and backward planning.

We explain the forward planning part in Section 4.1, and its two main modules, the BEV-planning module and video sense module, in Section 4.1.1 and Section 4.1.2, respectively. These parts show how trajectories are initially generated by leveraging multimodal data. In contrast to traditional ideas, we provide interpretability of behavior in the course of autonomous driving. Then, in Section 4.2, we explain backward planning, which aims to retrospectively review these strategies for potential risks, ensuring that safety is maintained throughout the decision-making process. In Section 6, we detail the experimental setup, highlighting SafeMod’s evaluation using the nuScenes and CARLA benchmarks.

In Section 7, we present the results and analysis, showing that SafeMod consistently outperforms other systems in terms of trajectory accuracy and collision avoidance. Experimental results on benchmarks like CARLA and nuScenes show that SafeMod outperforms state-of-the-art methods in navigation success rates, safety compliance, and overall performance. Its modular design also allows for seamless integration into various autonomous systems, offering flexibility for research and real-world applications.

In future work, we aim to improve SafeMod’s generalization by testing it in diverse environments (e.g., rural roads, highways, extreme weather) and incorporating multimodal data from additional sensors (e.g., LiDAR, radar). Efforts will also focus on enhancing computational efficiency for real-time operation. Incorporating reinforcement learning from human feedback (RLHF) could further refine the system by adapting navigation strategies to user preferences and human-like driving behaviors. Additionally, evaluating SafeMod’s scalability in fleet-level systems and traffic management is a key area for exploration.

In conclusion, SafeMod establishes a strong foundation for safe and intelligent autonomous navigation. Addressing these future challenges will enhance its adaptability, scalability, and real-world applicability, advancing the field of autonomous driving. 

## Figures and Tables

**Figure 1 sensors-24-06723-f001:**
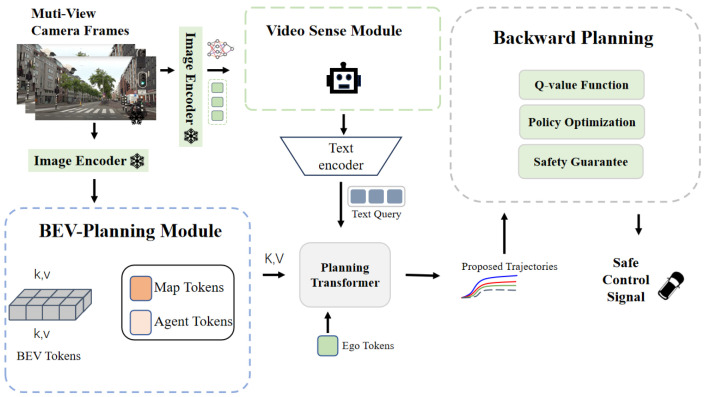
Overall framework of SafeMod. SafeMod takes multi-view image sequences as input, transforms them into BEV embedding and sense description, outputs them and samples one action to control the vehicle.

**Figure 2 sensors-24-06723-f002:**
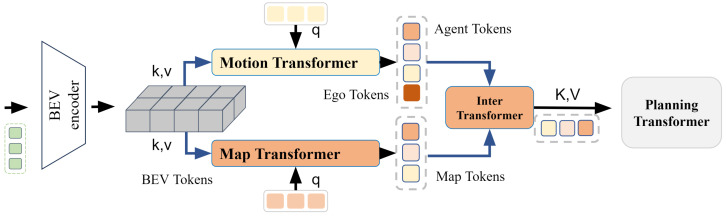
Framework detail of BEV-planning module. The BEV-planning module processes bird’s-eye-view (BEV) features through multiple transformers to generate strategic vehicle control actions, incorporating BEV feature extraction, motion and map predictions, inter-query interactions, and final trajectory planning based on high-level driving commands.

**Figure 3 sensors-24-06723-f003:**
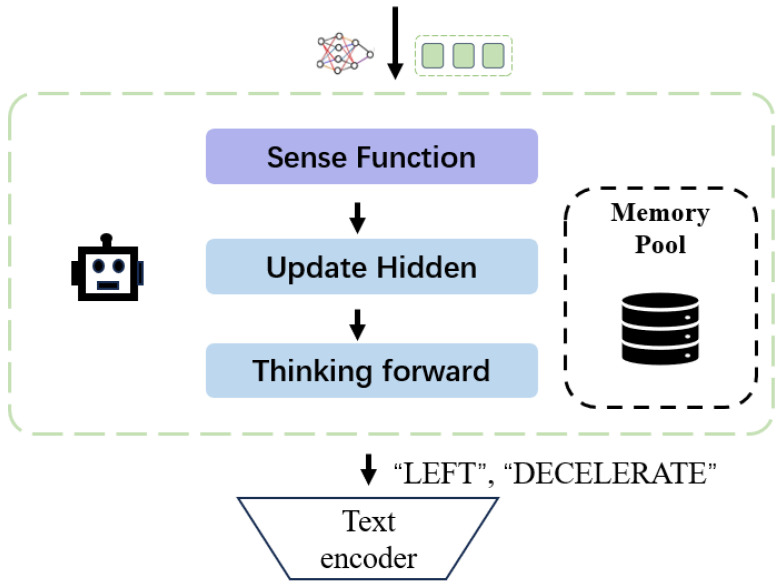
Overall framework of video sense module. The video sense module processes video inputs to extract structured sensory information and generate multi-turn dialogues for enhanced scene understanding, utilizing a pre-trained video encoder to obtain video embeddings, a cross-modality projector for alignment with language embeddings, and a perception module to continuously update the agent’s hidden states based on the extracted data, ultimately predicting context-specific actions for navigation.

**Figure 4 sensors-24-06723-f004:**
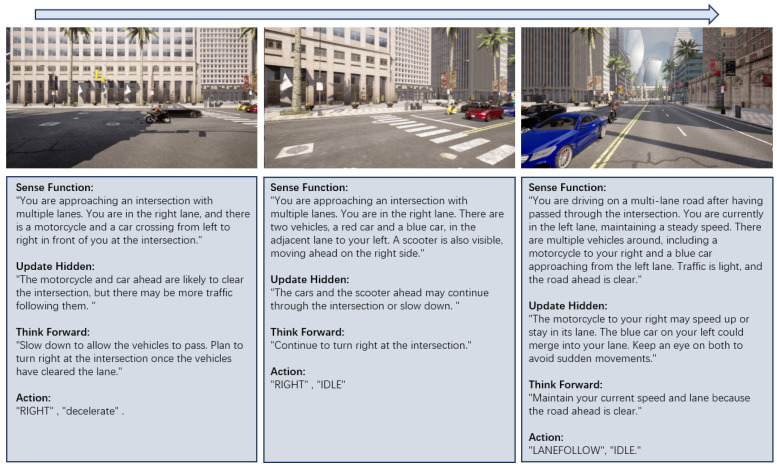
Intersection traffic negotiation. The module identifies the vehicles at the intersection, anticipates their behavior, and selects a safe right-turn maneuver while avoiding collisions. It analyzes the scooter and cars’ positions, predicting their paths and adjusting speed or direction to maintain safe distances. After passing the intersection, the framework continues monitoring for lane changes or sudden movements, ensuring smooth traffic flow and preventing conflicts.

**Figure 5 sensors-24-06723-f005:**
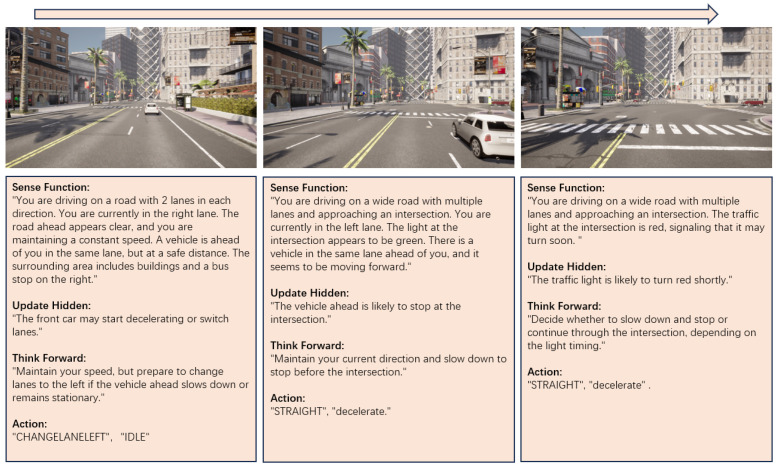
Lane changing. The module identifies the vehicle ahead while driving on a multi-lane road, anticipates its stationary behavior, and selects a safe lane change maneuver to the left. It analyzes the traffic light at the intersection and the presence of vehicles, predicting possible stops and adjusting speed accordingly to maintain safe movement.

**Figure 6 sensors-24-06723-f006:**
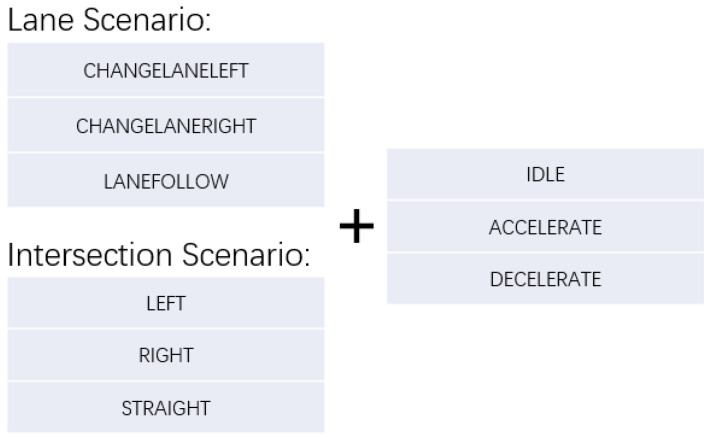
Example of action output.

**Figure 7 sensors-24-06723-f007:**
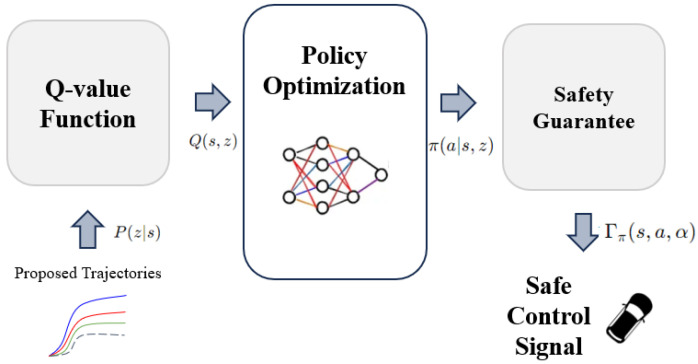
Policy generation in backward planning using Q-function optimization.

**Figure 8 sensors-24-06723-f008:**
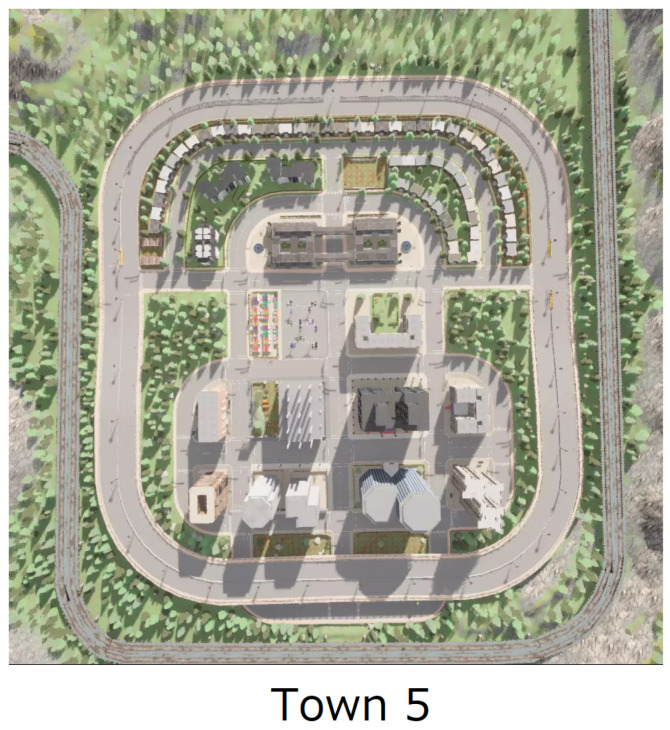
The aerial views of Town5 within the CARLA simulator. To evaluate the vehicle’s performance comprehensively, we used Town5 by focusing on several critical areas. First, it assesses the vehicle’s ability to manage right-of-way and prevent accidents at complex intersections involving multiple vehicles (traffic negotiation). Second, it evaluates the vehicle’s capacity to detect and circumvent suddenly appearing obstacles, such as road obstructions (obstacle avoidance). Third, it tests the vehicle’s responses to emergency stopping situations and its ability to execute swift lane changes to evade potential hazards (braking and lane changing).

**Figure 9 sensors-24-06723-f009:**
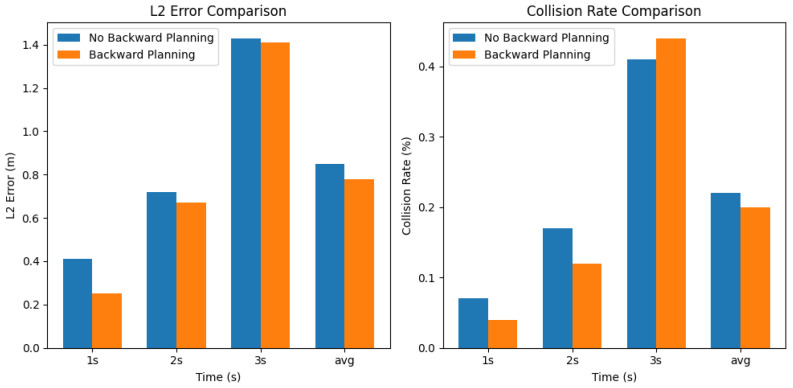
Backward performance test. The evaluation results indicate that the backward planning module consistently outperforms the baseline method across all metrics, especially in the 1 s interval of each metric. These improvements demonstrate the module’s capability to handle complex driving scenarios more safely and deliver more accurate autonomous driving.

**Table 1 sensors-24-06723-t001:** Open-loop planning performance. SafeMod demonstrates superior end-to-end planning effectiveness and maintains competitive inference speed on the nuScenes validation dataset. Notably, SafeMod integrates advanced decision-making strategies to optimize safety and responsiveness in dynamic environments. In open-loop evaluation, ego status information is disabled to ensure a fair comparison across methods.

Method	L2 (m) ↓				Collision (%) ↓			
	**1 s**	**2 s**	**3 s**	**Avg.**	**1 s**	**2 s**	**3 s**	**Avg.**
ST-P3 [36]	1.35	1.91	2.75	2.00	0.25	0.72	1.31	0.76
VAD [22]	0.31	0.79	1.52	0.87	0.06	0.15	0.48	0.23
FF [49]	0.56	1.21	2.56	1.44	0.09	0.21	1.09	0.46
EO [50]	0.62	1.41	2.42	1.48	0.06	0.17	1.12	0.45
UniAD [46]	0.51	0.98	1.71	1.07	0.07	0.13	0.74	0.31
GPT-Driver [48]	0.28	0.81	1.56	0.88	0.09	0.17	1.12	0.46
**SafeMod**	**0.25**	**0.67**	**1.41**	**0.78**	**0.04**	**0.12**	**0.44**	**0.20**

**Note:** Bold values indicate the best performance for each metric.

**Table 2 sensors-24-06723-t002:** Closed-loop simulation results. SafeMod achieves the best closed-loop planning performance on CARLA in the image input methods.

Method	Town05 Short		Town05 Long	
	**DS**↑	**RC**↑	**DS**↑	**RC**↑
CILRS [20]	7.43	13.47	3.71	7.21
Transfuser [47]	55.55	80.03	**32.17**	57.41
VAD [22]	65.32	88.14	31.01	74.94
ST-P3 [36]	54.88	86.32	11.04	**83.03**
**SafeMod**	**65.45**	**88.84**	32.02	78.66

**Note:** Bold values indicate the best performance for each metric.

**Table 3 sensors-24-06723-t003:** Isolated performance comparison of video sense module components.

Step			L2 (m) ↓				Collision (%) ↓			
**SF**	**UH**	**TF**	**1 s**	**2 s**	**3 s**	**Avg.**	**1 s**	**2 s**	**3 s**	**Avg.**
✓	-	-	0.68	0.91	1.62	1.07	0.44	0.62	0.90	0.65
✓	-	✓	0.44	0.75	1.46	0.77	0.12	0.23	0.44	0.26
✓	✓	✓	**0.41**	**0.69**	**1.41**	**0.71**	**0.07**	**0.17**	**0.41**	**0.22**

**Note:** Bold values indicate the best performance for each metric.

**Table 4 sensors-24-06723-t004:** Performance impact of video sense module on the SafeMod autonomous driving system.

System Configuration	Inference Latency (ms)	Frame Rate (FPS)
BP	50	20
BP + SF	75	20
BP + SF + UH + TF	92	19

## Data Availability

Access to the data in this study is temporarily restricted. The data are currently under review and will be available upon request from the corresponding author once the review is complete.

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
