# Peer review of "Bidirectional Planning for Autonomous Driving Framework with Large Language Model"

_sensors, 2024, doi:10.3390/s24206723_

Round 1

Reviewer 1 Report

Comments and Suggestions for Authors

This is a really interesting work, which not only creates a bidirectional planning structure with forward planning and backward planning, but also provides video sense module with LLM. I suggest a mionr review of this work due to its potential possibility for publication.

1. Too many description is given for experiments setup and evaluation metrics. Hence ,the Section 6 should be simplified, while the result analysis especially more details about closed-loop silmulation should be provided.

2. The part of LLM in Section 4.1.2 should be strengthened in accordance with the topic of this paper,especially how the LLM is implemented to predict surrounding agents’ intent invideo sense module . However, I think there is no really inovation for the adpoted method in backward planning such as SAC and CVAR, so this part may be overemphasised.   

3. The written and format of paper should be improved. 

4. The code of SafeMod is open-source?

Author Response

  I inadvertently sent the wrong file earlier. Please kindly refer to the latest version of the attachment.

Reviewer 2 Report

Comments and Suggestions for Authors

1. Since LLM and VLM added to the SafeMod model, does the real-time performance of the autonomous driving system be affected?

2. Reader will be confused the names of “Forward Planning” and “Backward Planning”, these names are somewhat reminder of the principles of “feedforward and feedback control”..

3. This article is too long. It is recommended to shorten some common knowledge (such as the content in Definition 1.1). It is also recommended to shorten many similar contents about each Transformer in “4.1.1. BEV-Planning Module”. Same recommendation to shorten the main content of Reinforcement Learning in 4.2.

4. In the “6 Experiments”, what’s the speed of autonomous driving vehicle? Does the vehicle speed maintain the same under various operating conditions? Does the vehicle speed affect the evaluation index?

5. In the experimental scenario, what types of obstacles are there in the collision analysis? Are there any dynamic moving obstacles?

6. According to Table 1, compared with some other methods, the improvement of the evaluation index ”L2 and Collision” in SafeMod is not much. The author can analyze the similarities of these methods with similar index value (for example, L2 index: VAD (0.31), GPT-Driver (0.28), SafeMod (0.25); the collision index values of some methods are basically the same).

Compared with these methods, in which specific scenarios does SafeMod method perform better? And what is the principle behind its better performance?

7. The author should do more analysis on the experimental results. For example, for specific scenarios, what improvements does “Bidirectional planning“ have compared to “Forward planning” or “Backward planning”? In some scenarios, the SafeMod method proposed in this paper is particularly useful; while in other scenarios, the improvement of this method is not much. Can the analysis be done on the mechanism level of the algorithm? Rather than just comparing the numerical values of the algorithm indicators.

Rather than just comparing the numerical values of different methods, should the author  analyze the simulation results from the algorithm mechanism level?

8.      There are some writing errors in the article, please check carefully and make corrections, such as the label "Inter Trsformer" in Figure 2.

    Line 195: "sense modules work in tandem", for autonomous vehicles, is the term "work in tandem" commonly used? Or using “collaborative work”?

Author Response

Please kindly refer to the latest version of the attachment.

Reviewer 3 Report

Comments and Suggestions for Authors

This paper introduces a novel autonomous driving framework that employs Large Language Models for bidirectional planning, enhancing safety and decision-making in complex environments. It demonstrates superior performance over existing methods through extensive experiments on CARLA and nuScenes benchmarks. This research is of some significance, but there are some issues that suggest additional improvements:

1.      When presenting related work, it may be beneficial to include a summary sentence that introduce the work.

2.      In Figure 8, please clearly introduce the figure, instead of “This environment provides a balanced challenge between high-speed driving and precision navigation.”.

3.      Please revise the conclusion, summarize the entire text concisely, and propose future work.

4.      Change sentences from passive to active where appropriate, e.g., instead of "The default numbers for BEV queries, map queries, and agent queries are 200 × 200, 100 × 20, and 300, respectively...." use "The default settings for BEV queries, map queries, and agent queries are set at 200 × 200, 100 × 20, and 300, respectively..."

Author Response

(The authors gave the same response as above.)
